# QDsim: An user-friendly toolbox for simulating large-scale quantum dot device

Valentina Gualtieri,[1] Charles Renshaw-Whitman,[1] Vinicius Hernandes,[1] and Eliska Greplova[1]

[1]*Kavli Institute of Nanoscience, Delft University of Technology, 2628 CJ Delft, the Netherlands*
(Dated: April 4, 2024)

We introduce QDsim, a python package tailored for the rapid generation of charge stability diagrams in large-scale quantum dot devices, extending beyond traditional double or triple dots. QDsim is founded on the constant interaction model from which we rephrase the task of finding the lowest energy charge configuration as a convex optimization problem. Therefore, we can leverage the existing package CVXPY, in combination with an appropriate powerful solver, for the convex optimization which streamlines the creation of stability diagrams and polytopes. Through multiple examples, we demonstrate how QDsim enables the generation of large-scale dataset that can serve a basis for the training of machine-learning models for automated tuning algorithms. While the package currently does not support quantum effects beyond the constant interaction model, QDsim is a tool that directly addresses the critical need for cost-effective and expeditious data acquisition for better tuning algorithms in order to accelerate the development of semiconductor quantum devices.

## INTRODUCTION

Quantum dots (QDs) have emerged as a particularly promising quantum computing platform [1–6]. These semiconducting systems, that operate by trapping charge carriers in potential wells called "dots", have been the subject of extensive research due to their scalability potential and the relative simplicity to fabricate them, that leverages already existing techniques in the semiconductor industry.

The scalability of quantum dot-based qubits is considered a cornerstone for practical quantum computation. At the same time, the complexity of these systems scales with the number of quantum dots, and it poses a significant challenge in establishing and maintaining the desired electron occupancy across the array.

Furthermore, with increasing device size, the task of manually tuning each quantum dot becomes impractical. The scaling of these systems necessitates an automated approach to tuning. In this context, artificial intelligence (AI) emerges as a highly promising solution. AI algorithms have the potential to learn and adapt to the complex variety of quantum dot behaviors, automating the tuning process with efficiency and precision.

However, the efficacy of AI depends on the availability of extensive datasets that capture the diverse operational regimes of quantum dot arrays. These datasets are critical for training robust machine learning. The scarcity of such data is a significant bottleneck in the advancement of AI applications within this field [7–16].

In response to these challenges, we present QDsim, a novel computational framework designed to simulate the electrostatic environment of quantum dot arrays efficiently. Our approach reduces the complexity of the simulation problem to a convex optimization task, offering an efficient and user-friendly solution. QDsim is implemented as a open-source Python package, providing a flexible tool for quantum dot array design and large-scale data generation for machine learning (ML) applications [17].

The most important feature of QDsim is its ability to generate charge stability diagrams, which are essential for understanding the operational regimes of quantum dot arrays. These diagrams show the connection between gate voltages and charge configurations, and are characterized by a tessellation of the voltage space into polytopes. The geometry of these polytopes provides insights into the charge configuration of the quantum dot system. While other simulators of quantum dot arrays exist, QDsim package offers unprecedented flexibility in geometry of the device in term of placement of dots, gates and sensors. Additionally, we demonstrate a significant speed in the charge stability diagram generation that allows for rapid simulations of 100+ quantum dots.

By enabling the rapid generation of charge stability diagrams for large-scale quantum dot devices, QDsim serves as a foundational tool for creating the vast datasets required for machine learning training. Its ability to simulate complex quantum dot arrays and produce charge stability diagrams is a step towards the future where AI-driven automatization becomes the standard for quantum device tuning.

The paper is structured as follows. In Section  we introduce the theory behind the model: constant capacitance model and Coulomb polytopes. In Section  we discuss key QDsim classes and provide detailed explanation of the package functionalities. In Section we discuss relevant examples for the QDsim. We provide detailed discussion of default device designs and their possible customization as well as instructions on how to a design and simulate completely custom architectures.

## FORMULATION OF THE ELECTROSTATIC MODEL: CHARGE STABILITY DIAGRAMS

In this section, we describe the theoretical foundation of the QDsim package: the constant-interaction (or

constant-capacitance) model applied to quantum dot arrays.

Charge stability diagrams are visual manifestations of the Coulomb blockade effect, a quantum phenomenon that occurs in small conducting or semiconducting structures, such as quantum dots. At the quantum scale, electrons are not free to flow into and out of a quantum dot without restriction; instead, they are influenced by the Coulomb force from other electrons within the dot. When an electron is added to a quantum dot, it increases the energy of the system due to this repulsive force. If the energy required to add another electron exceeds the thermal energy of the system, the dot will not take on any additional electrons until the external conditions (such as gate voltage) are altered. This leads to a blockade of charge transfer, which is observable as discrete jumps in the conductance through the quantum dot.

Experimentally, charge stability diagrams are obtained by varying the voltages on the electrostatic gates that control the quantum dots and measuring the resulting conductance. For multiple dots devices, these diagrams exhibit a characteristic pattern, each corresponding to a stable number of electrons within the quantum dots. The boundaries between these regions represent points of charge degeneracy, where the number of electrons on a dot can change.

In our model, we represent these phenomena within the classical framework of the constant interaction [18]. In this model, each quantum dot is considered a conductor with capacitive coupling to other dots and to electrostatic gates. The charge stability diagrams emerge from this model as a tessellation of the gate voltage space, where each region corresponds to a stable charge configuration. These configurations are the ground states of the system's free energy function.

By simulating charge-stability diagrams, QDsim provides a powerful tool for efficiently simulating the behavior of quantum dot arrays.

### Derivation of the Constant-Capacitance Model Energy Equation

In the constant interaction model, each quantum dot is considered a conductor with capacitive coupling to other dots and to electrostatic gates [19] [20].

The electrostatic characteristics of the system with $N_D$ dots and $N_G$ gates are captured by two mutual capacitances matrices:

- the dot-to-dot mutual capacitance matrix, in which each element $\widetilde{C_{ij}^{DD}}$ represents the capacitive coupling between dot $i$ and dot $j$, and the diagonal elements represent the dot's self capacitance; and

- the dot-to-gate mutual capacitance matrix, in

which each element $\widetilde{C_{ij}^{DG}}$ represents the capacitive coupling between dot $i$ and gate $j$.

The dot-to-dot mutual capacitance matrix $\widetilde{C^{DD}}$ requires $N_D(N_D - 1)/2$ values due to the symmetrical relation of the mutual capacitances, i.e. the capacitance between element $i$ and element $j$ is the same between element $j$ and element $i$. No symmetry relations are present in the dot-to-gate mutual capacitance matrix $\widetilde{C^{DG}}$, therefore it is defined by $N_D N_G$ values.

Capacitance, $C$, relates the charge state $Q$, i.e. the charge contained on the conductor, to the electrostatic potential, $V$ via $Q = CV$. For multiple conductors, $Q$ and $V$ are column vectors $\mathbf{Q}$ and $\mathbf{V}$, and $C$ is a matrix $\mathbf{C}$. However, in order for the relation

$$\mathbf{Q} = \mathbf{CV} \qquad (1)$$

to hold true in matrix form, some distinctions must be made. Here we first introduce the total system's mutual capacitance $\widetilde{C}$, which is obtained by stacking the dot-to-dot and dot-to-gate capacitance matrices in the following way:

$$\widetilde{\mathbf{C}} = \begin{bmatrix} \widetilde{\mathbf{C}}_{DD} & \widetilde{\mathbf{C}}_{DG} \\ \widetilde{\mathbf{C}}_{DG}^T & \mathbb{1} \end{bmatrix}. \qquad (2)$$

Matrix $\widetilde{\mathbf{C}}$ satisfies the condition

$$Q_i = \sum_j \widetilde{C_{ij}}(V_i - V_j), \qquad (3)$$

which is different from the aforementioned $\mathbf{Q} = \mathbf{CV}$. Here the indexes $i$, $j$ run from 1 to $N_D + N_G$. $V_i$ represents the voltage applied to the $i$-conductor. $V_j$ represents the voltage applied to the $j$-conductor. These conductors are both dots and gates, accounting for a total of $N_D + N_G$ conductors. Specifically, in our notation, a general index $i$ running from 1 to $N_D$ will account for the dots, while the same index running from $N_D + 1$ to $N_D + N_G$ will account for the gates.

Upon manipulation, the relation $\mathbf{Q} = \mathbf{CV}$ holds when $\mathbf{C}$ is the Maxwell matrix [21], defined as:

$$C_{ij} = \delta_{ij} \sum_k \widetilde{C}_{ik} + (1 - \delta_{ij})(-\widetilde{C}_{ij}), \qquad (4)$$

where $\delta_{ij}$ is the Kronecker delta.

Given that all self-capacitances $\widetilde{C}_{ii}$ are positive, the Maxwell matrix is strictly diagonally dominant [22], ensuring its invertibility via the Levy-Desplanques theorem [23].

Distinguishing between dot and gate properties, we express $\mathbf{Q}$ , $\mathbf{V}$ and $\mathbf{C}$ as:

$$\mathbf{Q} = \begin{bmatrix} \mathbf{Q}_D \\ \mathbf{Q}_G \end{bmatrix}, \quad \mathbf{V} = \begin{bmatrix} \mathbf{V}_D \\ \mathbf{V}_G \end{bmatrix}, \quad \mathbf{C} = \begin{bmatrix} \mathbf{C}_{DD} & \mathbf{C}_{DG} \\ \mathbf{C}_{DG}^T & \mathbf{C}_{GG} \end{bmatrix}. \qquad (5)$$

The capacitance relation in Equation (1) then becomes:

$$\begin{bmatrix} \mathbf{Q}_D \\ \mathbf{Q}_G \end{bmatrix} = \begin{bmatrix} \mathbf{C}_{DD} & \mathbf{C}_{DG} \\ \mathbf{C}_{DG}^T & \mathbf{C}_{GG} \end{bmatrix} \begin{bmatrix} \mathbf{V}_D \\ \mathbf{V}_G \end{bmatrix}. \quad (6)$$

The free energy function $F$ of the system is given by:

$$F = U - W = \frac{1}{2} \left[ \mathbf{Q}_D^T, \mathbf{Q}_G^T \right] \begin{bmatrix} \mathbf{V}_D \\ \mathbf{V}_G \end{bmatrix} - \mathbf{V}_G^T \mathbf{Q}_G. \quad (7)$$

Assuming $\mathbf{Q}_D = e\mathbf{N}_D$, where $e$ is the elementary charge with sign ($+1$ for holes, $-1$ for electrons), we can rewrite the free energy function in terms of $\mathbf{N}_D$ (the charge configuration that we want to obtain) and $\mathbf{V}_G$ (the gate voltages that we want to tune). Omitting gate self-energies, we rewrite the free energy from Equation (7) as

$$\begin{aligned} F\left(\mathbf{N}_D; \mathbf{V}_G\right) = &\frac{1}{2} \left( e^2 \mathbf{N}_D^T \mathbf{C}_{DD}^{-1} \mathbf{N}_D \right. \\ &- 2e\mathbf{N}_D^T \mathbf{C}_{DD}^{-1} \mathbf{C}_{DG} \mathbf{V}_G \\ &+ \left. \mathbf{V}_G^T \mathbf{C}_{DG}^T \mathbf{C}_{DD}^{-1} \mathbf{C}_{DG} \mathbf{V}_G \right). \end{aligned} \quad (8)$$

For the remainder of the paper, we will adopt units such that $|e| = 1$, and define the charging-energy matrix $\mathbf{E}_C = \mathbf{C}_{DD}^{-1}$. It is worth noting that we will need to take into account the sign of the charge, which will be negative for electrons, and positive for holes simulations.

### Ground States and Coulomb Polytopes in V-Space

In this section we show that the ground states of the free energy function define the polytopes in the voltage space. We define the ground-state energy $F_{GS}(V_G)$ and the corresponding occupation numbers $GS(V_G)$ as

$$F_{GS}(\mathbf{V}_G) = \min_{\mathbf{N} \in \mathbb{Z}^{N_D}} F(\mathbf{N}; \mathbf{V}_G), \quad (9)$$

$$GS(\mathbf{V}_G) = \{ \mathbf{N} \in \mathbb{Z}^{N_D} \mid F(\mathbf{N}, \mathbf{V}_G) = F_{GS}(\mathbf{V}_G) \}. \quad (10)$$

The set of voltages for which a given occupation $\mathbf{N}_0$ is a ground state, $GS^{-1}(\mathbf{N}_0)$, is determined by the condition that $F(\mathbf{N}_0, \mathbf{V}_G) \leq F(\mathbf{N}_0 + t, \mathbf{V}_G)$ for all $t \in \mathbb{Z}^{N_D}$. This leads to the the following description of a convex polytope

$$(t^T \mathbf{E}_C \mathbf{C}_{DG})\mathbf{V}_G \preceq \frac{1}{2} t^T \mathbf{E}_C t + t^T \mathbf{E}_C \mathbf{N}_0, \quad (11)$$

where $\preceq$ denotes element-wise inequality.

Convex polytopes can be defined as an intersection of a finite number of half-spaces. This definition is called a half-space representation or H-description [24]. Inequality (11) represents the H-description of the coulomb polytopes.

Direct consequence of the inequality (11) is that the regions in $V_G$-space admitting a particular occupation $\mathbf{N}$

as a ground state form convex polytopes. We can also prove that two polytopes sharing an interior point $\mathbf{V}_0$ must coincide, implying that two states $\mathbf{N}_1$ and $\mathbf{N}_2$ are degenerate if and only if $\mathbf{N}_1 - \mathbf{N}_2 \in \text{Null}(\mathbf{C}_{DG}^T \mathbf{E}_C)$.

To prove this statement, we can consider two polytopes which share a point $\mathbf{V}_0$ interior to each, therefore all the inequalities in (11) hold strictly. Suppose that two ground-states, $\mathbf{N}_1$ and $\mathbf{N}_2$, are admitted for the point $\mathbf{V}_0$, which is in the interior of the respective polytopes. Then there exists an open ball $B$ of finite radius $\eta \in \mathbb{R}^{++}$ such that all the points in $B_\eta(\mathbf{V}_0)$ are also in both polytopes. Within this ball, by assumption the occupations $\mathbf{N}_1$ and $\mathbf{N}_2$ remain ground states, therefore have the same energy. Then for all $\delta \mathbf{V} \in B_\eta(0)$, the following holds:

$$F(\mathbf{N}_1, \mathbf{V}_0 + \delta \mathbf{V}) = F(\mathbf{N}_2, \mathbf{V}_0 + \delta \mathbf{V}) \quad (12)$$

$$\begin{aligned} \frac{1}{2}(\mathbf{N}_1^T \mathbf{E}_C \mathbf{N}_1 - \mathbf{N}_2^T \mathbf{E}_C \mathbf{N}_2) - (\mathbf{C}_{DG}\mathbf{V}_0)^T \mathbf{E}_C (\mathbf{N}_1 - \mathbf{N}_2) \\ = (\mathbf{C}_{DG}\delta \mathbf{V})^T \mathbf{E}_C (\mathbf{N}_1 - \mathbf{N}_2) \quad (13) \end{aligned}$$

The left hand side is equal to 0 when $\delta \mathbf{V} = 0$. Hence, the right-hand side vanishes independently of $\delta \mathbf{V}$. Thus $N_1^T \mathbf{E}_C \mathbf{C}_{DG} = N_2^T \mathbf{E}_C \mathbf{C}_{DG}$. It follows that $F(\mathbf{N}_1, \mathbf{V}_0) = F(\mathbf{N}_2, \mathbf{V}_0)$ for all $\mathbf{V}_0 \in \mathbb{R}^{N_G}$. Further, any two states are degenerate if and only if $\mathbf{N}_1 - \mathbf{N}_2 \in \text{Null}(\mathbf{C}_{DG}^T \mathbf{E}_C)$ [25].

In this section, we have demonstrated that the task of identifying Coulomb diamonds can be reformulated through convex optimization. The established convexity enables the framing of Equation (8)'s minimization as a convex problem. This approach leads to the determination of ground states as the sought-after solution.

### QDSIM PACKAGE

QDsim is a Python package that bridges the gap between theoretical frameworks and practical quantum dot device simulations. This versatile tool comprises three essential classes: QDDevice (quantum dot device), QDSimulator (quantum dot simulator), and CapacitanceQuantumDotArray (capacitance quantum dot array). All the code is available in a public repository on GitLab [26].

### The quantum dot device class: QDDevice

The QDDevice class is a key element for device definition. It focuses on translating design of the device into capacitance matrices, specifically addressing dot-to-dot and dot-to-gate mutual capacitance matrices. This class is responsible for defining key parameters related to the device's geometry and design. Users can specify

the number of dots, the number of gates, the locations of the dots, as well as the dot-to-dot $\widetilde{\mathbf{C}_{DD}}$ and dot-to-gate $\widetilde{\mathbf{C}_{DG}}$ mutual capacitance matrices.

Upon initialization, the `QDDevice` class provides an empty object, which users can then populate with the desired device characteristics. A range of standard design options is available, each tailored to specific device configurations. These standard options include the following methods:

- `one_dimensional_dots_array`: This option configures a line of dots with individual gate control, where users can define the number of dots, their locations, the dots' self-capacitance, the average dot-to-gate capacitance, if the dots are equal (i.e. they all have the same self-capacitance), if the gates are equal (i.e. if the dot-to-gate capacitance is the same for every couple in which the gate directly controls the dot), and the strength of the crosstalk interaction.

- `bi_dimensional_10_dots_array` : It sets up a 2D array of 10 dots with individual gate control, allowing customization of the device properties and capacitances as listed in the previous case.

- `crossbar_array_shared_control`: This option creates a crossbar array of dots with shared control. Users can specify the number of dots per side of the square lattice, and other device characteristics as in the previous case.

The `QDDevice` class features several attributes, including device type (for plotting purposes), the number of dots, the number of gates, dot self-capacitance, dot locations, dot-to-dot mutual capacitance matrices, and dot-to-gate mutual capacitance matrices.

Additionally, `QDDevice` offers methods for setting dot locations, the number of gates, custom dot-to-dot mutual capacitance matrices, and custom dot-to-gate mutual capacitance matrices. It also provides functionality for automatically evaluating dot-to-dot and dot-to-gate mutual capacitance matrices based on dot locations, assuming individual gate control. These methods facilitate the definition and customization of quantum dot devices and enable users to configure the device to their specific requirements, making it a versatile tool for simulations. Here, we provide an overview of the key methods offered:

- `set_physical_dot_locations`: This method allows users to assign dot locations to the `QDDevice` object. By specifying the coordinates $(x, y)$ of each dot in the device, users can precisely define the spatial arrangement of quantum dots.

- `set_dot_dot_mutual_capacitance_matrix`: With this method, users can assign a custom dot-to-dot mutual capacitance matrix to the `QDDevice`

object. This level of customization enables precise modeling of the capacitance interactions between quantum dots.

- `set_dot_gate_mutual_capacitance_matrix`: Similarly, users can define a custom dot-to-gate mutual capacitance matrix using this method. The dot-to-gate capacitance matrix plays a crucial role in simulating the interactions between quantum dots and gate electrodes.

- `evaluate_dot_dot_mutual_capacitance_matrix`: This method calculates the dot-to-dot mutual capacitance matrix of the `QDDevice` based on the dot locations. It employs a distance-based model to compute the capacitance interactions, taking into account the arrangement of quantum dots.

- `evaluate_dot_gate_mutual_capacitance_matrix`: This method is used to compute the dot-to-gate mutual capacitance matrix based on the dot locations. It assumes that each gate corresponds to a dot in only controls that dot, making it suitable for certain device configurations.

The package's versatility extends to device visualization, with a plotting method `plot_device` that allows users to create graphical representations of the device, including optional sensors and dot labels. The generated plots are exportable in various image formats, such as PDF and PNG, allowing flexibility in saving visual representations. Device attributes can also be exported to JSON files with the `save_to_json` method, providing a structured format for documentation. These JSON files can be easily imported and utilized using the `load_from_json` class method.

### *Indexing*

We begin by explaining the indexing conventions used for `one_dimensional_dots_array`, `crossbar_array_shared_control`, and potential custom configurations.

In the context of `QDsim` package, each dot and gate is associated with a distinct index, designated as $i$ and $j$ respectively. These indices range from 0 to $N_D - 1$ for dots and $N_G - 1$ for gates. For instance, the matrix element $\widetilde{\mathbf{C}}_{ij}^{DD}$ denotes the mutual capacitance between dot $i$ and dot $j$. This notation is consistently applied across the various mutual capacitance matrices within the package.

In the standard design templates provided, the indexing scheme of the dots and gates is readily represented through a plot of the device, via the `plot_device` method. Within the `one_dimensional_dots_array` configuration, the dots and their corresponding gates

are sequentially numbered from left to right, from 0 to $N_D - 1$. Given the one-to-one correspondence between dots and gates in this arrangement, gate indices are not explicitly depicted; however, they adhere to the same left-to-right numbering convention, extending from 0 to $N_G - 1$, where $N_G$ equals $N_D$ in this case.

The `crossbar_array_shared_control` design has a distinct indexing pattern, as can be seen in Figure 1.

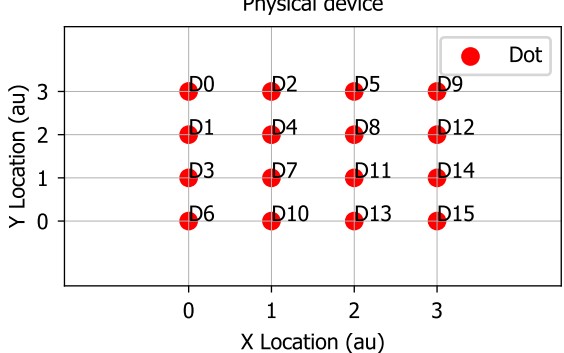

FIG. 1. Example of the indexing of the dots in the crossbar array design. The 'D' stands for 'dot', the adjacent number represents the index.

Here, dots are numbered starting from the upper left corner, progressing downwards, and subsequently ascending diagonally towards the right. This process is repeated, descending from the leftmost point for one step and ascending in the rightward diagonal direction.

This convention ensures that increasing dot indices are governed by the same gate. The gates themselves are arranged diagonally and are numbered from the upper left to the lower right corner.

It is of course possible for users to modify our template configurations, but when doing so it is essential to adhere to the above described indexing convention such that correct capacitance matrices are generated.

### The simulator class: `QDSimulator`

The `QDSimulator` class is a fundamental component of the `QDsim` package, and serves as the user's interface for simulating quantum dot charge stability diagrams.

The `QDSimulator` class is built atop the `CapacitanceQuantumDotArray` class. The `CapacitanceQuantumDotArray` class is utilized to construct and solve the convex optimization problem that characterize the quantum dot array. It will be covered in details in the next section. At this point, the important distinction to be made is that the `QDSimulator` class is a wrapper of the `CapacitanceQuantumDotArray` class. Therefore the user will interact directly only with the `QDSimulator`

class, leaving the `CapacitanceQuantumDotArray` class working in the background.

When creating an instance of `QDSimulator`, the user has the discretion to specify the type of simulation—either 'Electrons' or 'Holes'. By default, the system assumes an 'Electrons' simulation.

The key attributes of the `QDSimulator` class are the following:

- `_qd_device`: This attribute represents the quantum dot device to be simulated and must be an instance of the `QDDevice` class.

- `_physics_to_be_simulated`: Determination whether the device consists of 'Electrons' or 'Holes'.

- `_variable_gate_index_1` and `_variable_gate_index_2`: These represent the indices of the gates to be scanned, i.e. which gates will be shown in the charge stability diagram axes. For simulations of the charge stability diagram, only a pair of gates can be simultaneously scanned.

- `_voltage_ranges`: This attribute sets the minimum and maximum voltages for the x- and y-axes, associated with gates indexed by `_variable_gate_index_1` and `_variable_gate_index_2`, respectively.

- `_sensor_locations`: This attribute determines the spatial coordinates (x, y) of each sensor within the device layout, which must be set via the class method `set_sensor_locations`. While the simulation itself, framed as a minimization problem, is independent of sensors, they are incorporated for more realistic visualization purposes. Specifically sensors allow visualization of realistic potential and current values and they would be monitored in an experiment.

The core method of this class is`simulate_charge_stability_diagram`. Central to the `QDsim` package, this function bridges the representation of the quantum dot device, defined by the `QDDevice` object, to the framework of the constant interaction model. The method utilizes a `CapacitanceQuantumDotArray` object, which serves as the powerhouse driving the entire simulation.

Users can choose their preferred convex optimization solver, including options like the open-source SCIP or the licensed MOSEK, for use within CVXPY. This higher-level interface seamlessly integrates with either MOSEK or SCIP as its back-end solvers, offering flexibility in solver selection.

Following the philosophy of flexibility, this method allows the specification of the gates to be scanned, with the number of probe points on both x- and y-axes, and an individual voltage range for each. Furthermore, the fixed voltage approach ensures a fixed potential for non-scanned gates, while the gates' voltages can be individually defined for more granular control. All the details will be described in Section by taking advantage of some use cases.

A beneficial feature is the saving mechanism: The resulting arrays of occupation, potential, and current can be stored using the associated file path parameters. This ensures that simulation data can be revisited or shared without the need to rerun computations.

The final feature of the `QDSimulator` class is its inbuilt capability to visualize the simulated results. Through its integrated plotting methods, users can render charge stability diagrams.

The `plot_charge_stability_diagrams` method of the class is designed to create a visual representation of the charge stability diagram. Its features include:

- **Colormap Customization**: By default, the method employs the 'RdPu' colormap. However, users can modify the colormap using the `cmapvalue` argument.

- **Noise Inclusion**: The method allows for the introduction of Gaussian, white, or pink noise to the plots, enhancing the realism of simulations. They can be added by using the boolean `gaussian_noise`, `white_noise`, and `pink_noise` arguments.

- **Potential vs. Current Mapping**: While the default visualization mode displays the current map, there exists an option to showcase the potential map instead by setting the boolean argument `plot_potential` to True.

- **Custom Noise Parameters**: Users have the liberty to specify parameters for Gaussian, white, and pink noise using the `gaussian_noise_custom_params`, `white_noise_custom_params`, and `pink_noise_custom_params` arguments. In the absence of user-defined values, default settings are applied.

- **Saving Plots**: If desired, the generated visualizations can be saved to a predetermined file path, set via the `save_plot_to_filepath` argument.

## The powerhouse of the package: the `CapacitanceQuantumDotArray` class

The `CapacitanceQuantumDotArray` class acts as the core computational engine of our framework. It completes the task of defining the problem parameters, establishing the correct environment, and running the actual simulation. This class operates predominantly in the background; users typically interact with higher-level interfaces such as `QDDevice` and `QDSimulator` and do not directly engage with `CapacitanceQuantumDotArray`.

The `CapacitanceQuantumDotArray` class is rooted in convex optimization techniques, particularly leveraging the CVXPY package. It aims to minimize the system's free energy, defined in Equation 7.

To obtain the system's ground state defined in Equation 9, we manipulated the free energy expression (Equation 8) in terms of $\mathbf{N}_D$ and $\mathbf{V}_G$, where $\mathbf{N}_D$ delineates the dot occupations. The free energy undergoes minimization concerning $\mathbf{N}_D$, ensuring that $\mathbf{N}_D$ remains an integer vector.

For simplicity in calculations and units, the class assumes the unit charge $|e|$ as 1, meaning the free energy is denoted in eV.

Here we present a brief overwiew of the key methods:

- `select_solver`: This method allows to choose a solver for the convex optimization problem. Available options include 'MOSEK' and 'SCIP'. This method takes in input the solver selected by the user while interacting with the `QDSimulator` class. The user never access the methods of the `CapacitanceQuantumDotArray` class directly.

- `probe_voltage_space`: It explores the entire voltage space and determines the ground state for each point, returning both the dot occupations and associated energy.

- `_find_ground_state`: For a specific voltage point, this method identifies the system's ground state.

- `_set_up_convex_optimization_problem`: This method prepares the convex optimization problem, serving as a foundation for `_find_ground_state`.

- `_evaluate_maxwell_matrices`: Here, the system's Maxwell capacitance matrix is determined, acting as a precursor for the optimization setup.

## EXAMPLES

In this section we highlight several common use-cases of QDsim, some of which corresponding to recently experimentally achieved devices [6] [5].

## The double dot device

Let us begin with a fundamental example of quantum dot device: double quantum dot. We begin with a `QDDevice` object in order to specify the physical parameters of the device. The `one_dimensional_dots_array` method significantly streamlines this initialization. By setting the `n_dots` parameter to 2, the system is automatically configured as a double dot device. The physical parameters of this default configuration can be seen by calling the `print_device_info` method, as shown below.

**Python Code**

```python
from qdsim import QDDevice, QDSimulator

# create a quantum dot device object
qddevice = QDDevice()
# double dot
qddevice.one_dimensional_dots_array(
    n_dots=2)
# print the device information
qddevice.print_device_info()
```

The output generated is as follows:

**Code Output**

Device type: in-line array
Number of dots: 2
Number of gates: 2
Physical dot locations: $[(0,0),(1,0)]$
Dot-dot mutual capacitance matrix:

$$\begin{bmatrix} 0.12 & 0.08 \\ 0.08 & 0.12 \end{bmatrix}$$

Dot-gate mutual capacitance matrix:

$$\begin{bmatrix} 0.12 & 0.00 \\ 0.00 & 0.12 \end{bmatrix}$$

Upon specifying the `n_dots`, the `QDDevice` class autonomously initializes all requisite attributes. However, the package offers flexibility for further customization, either through standard alteration functions or by manual attribute assignment using setter methods.

The following code snippets illustrate both methods of customization.

*Customization via default attributes*

In this double-dot scenario, we leverage the built-in modification functions accessible through the architecture specification method. This is achieved by specifying

`equal_dots = False`, `equal_gates = False`, and/or adjusting the `crosstalk_strength` parameter within a range from 0 (indicating no crosstalk) to 1 (representing the maximum threshold of crosstalk, determined in proportion to the capacitances within the simulation). For users seeking further customization, the capacitance values can be adjusted by examining and modifying the source code as necessary.

**Python Code**

```python
from qdsim import QDDevice, QDSimulator

# create a quantum dot device object
qddevice = QDDevice()
# double dot
qddevice.one_dimensional_dots_array(
    n_dots=2, equal_dots=False,
    equal_gates=False,
    crosstalk_strength=0.3)
# print the device information
qddevice.print_device_info()
```

The output generated is as follows:

**Code Output**

Device type: in-line array
Number of dots: 2
Number of gates: 2
Physical dot locations: $[(0,0),(1,0)]$
Dot-dot mutual capacitance matrix:

$$\begin{bmatrix} 0.12 & 0.08 \\ 0.08 & 0.11 \end{bmatrix}$$

Dot-gate mutual capacitance matrix:

$$\begin{bmatrix} 0.13 & 0.02 \\ 0.02 & 0.15 \end{bmatrix}$$

In this example, the alteration in the dot-to-dot and dot-to-gate mutual capacitance matrices is achieved by the introduction of random values. Using `equal_dots = False` will add random values on the diagonal of the dot-to-dot mutual capacitance matrix, while `equal_gates = False` will add random values to the diagonal of the dot-to-gate mutual capacitance matrix. Setting a value for `crosstalk_strength` will add random numbers to the off-diagonal, to ensure crosstalk effects. The random values applied can be both positive and negative, and are properly scaled with respect to the order of magnitude used in the matrices to ensure the maintenance of realistic and physically plausible parameters, i.e. they would only account for small variations of the values, roughly

10-20% variations.

*Customization via setter methods*

Conversely, for users seeking to employ custom capacitance matrices, the package provides two setter methods for this purpose: `set_dot_dot_mutual_capacitance_matrix` and `set_dot_gate_mutual_capacitance_matrix`.

```python
from qdsim import QDDevice, QDSimulator

# create a quantum dot device object
qddevice = QDDevice()

# double dot
qddevice.one_dimensional_dots_array(
    n_dots=2)

# define the custom capacitance matrices
cdd = np.array([[0.10, 0.7],[0.7, 0.15]])
cdg = np.array([[0.14, 0.3],[0.3, 0.12]])

# modify the class attributes
qddevice.
    set_dot_dot_mutual_capacitance_matrix(
        cdd)
qddevice.
    set_dot_gate_mutual_capacitance_matrix(
        cdg)

# print the device information
qddevice.print_device_info()
```

The output generated is as follows:

**Code Output**

Device type: in-line array
Number of dots: 2
Number of gates: 2
Physical dot locations: $[(0,0),(1,0)]$
Dot-dot mutual capacitance matrix:

$$\begin{bmatrix} 0.10 & 0.07 \\ 0.07 & 0.15 \end{bmatrix}$$

Dot-gate mutual capacitance matrix:

$$\begin{bmatrix} 0.14 & 0.03 \\ 0.03 & 0.12 \end{bmatrix}$$

When customising the capacitance matrices, it is key to pay attention to the indexing convention described in Section and to guarantee the symmetry of the dot-to-dot mutual capacitance matrix.

We recommend to use `plot_device` plotting method to verify that dots and gates are ordered as intended.

The inclusion of a sensor (along with its label) in the plot for enhanced visualization, as well as the capability to export the plot to a file with a preferred format, is achieved by executing the line of code provided below.

```python
# plot the device, the sensor
# and save the plot to a file
qddevice.plot_device(
    sensor_locations=[[2,1]],
    sensor_labels=['S0'],
    save_plot_to_filepath='dqd_device.pdf')
```

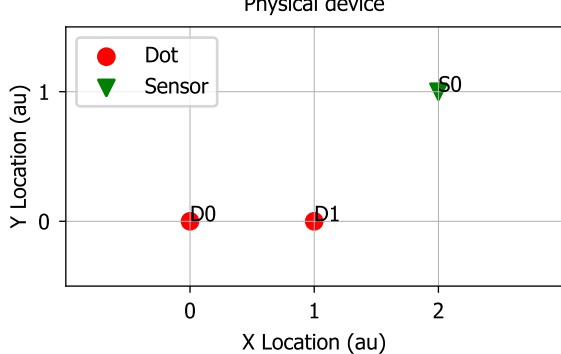

FIG. 2. A schematic plot of the double quantum dot architecture with sensor, which is the output of the `plot_device` method.

It is pertinent to note that the plotting method does not render the gates. This omission is intentional for two primary reasons: firstly, the geometry of the gates is not critical for simulation objectives, as the interactions between gates and dots are encapsulated within the dot-to-gate mutual capacitance matrices. Secondly, in devices with individual control mechanisms, the indexing for gates and dots is identical. We provide a schematic depiction of gates only for the cross-bar device, where the shared control aspect (single gate controls multiple dots) requires an indexing guidance.

*Simulation*

To start the simulation of the device, create an instance of the `QDSimulator` class. This class has a method named

simulate_charge_stability_diagram, which accepts qd_device as one of its parameters. Thus, an instance of the QDDevice class designated for simulation is inputted into the QDSimulator's simulation method, rendering the simulator class agnostic to the specific device being simulated. Consequently, the code snippets provided herein are applicable across all use-cases discussed within this Examples section.

```python
# create a quantum dot simulator object
# simulating electrons
qdsimulator = QDSimulator(simulate=
                                'Electrons')

# set the sensor locations
qdsimulator.set_sensor_locations([[2, 1]])

# simulate the charge stability diagram
qdsimulator.
    simulate_charge_stability_diagram(
        qd_device=qddevice, solver='MOSEK',
        v_range_x=[-5, 20],
        v_range_y=[-5, 20],
        n_points_per_axis=60,
        scanning_gate_indexes=[0, 1],
        use_ray=True)
```

After initializing a QDSimulator object with the intention of simulating an electron-based quantum dot device, the simulate attribute is set to 'Electrons'. This attribute can alternatively be configured to 'Holes'. If this attribute remains unspecified, the simulation defaults to 'Electrons'.

The subsequent step is determination of the sensor locations within the simulation environment. This is achieved by specifying their coordinates in Cartesian format $[[x_0, y_0], [x_1, y_1], \ldots]$ through the set_sensor_locations method.

The simulation process is then executed via the simulate_charge_stability_diagram method, requiring the specification of several parameters. These include the qd_device parameter, which requires an instance of the QDDevice class, and the selection of an optimization solver ('MOSEK' for licensed use or 'SCIP' for an open-source option via CVXPY). When unspecified, the solver defaults to 'SCIP'. Voltage ranges along the x and y axes are defined by v_range_x and v_range_y, respectively, with n_points_per_axis determining the plot's resolution. The indices of the gates to be scanned are specified in scanning_gate_indexes, with a maximum of two gates allowed simultaneously. The indexing order is significant, as the first index always denotes the x-axis and the second the y-axis. Additionally, the attribute use_ray=True can

be employed to leverage the Ray parallelization library [27] [28] for enhanced computational efficiency. By using Ray we can speed up the computational time by parallelizing the computation at each pixel of the plot.

*Plotting and adding noise*

The plot_charge_stability_diagrams method visualizes the simulation outcomes, plotting either the potential or current landscape, with an option to incorporate noise. To visualize the sensed potential, the plot_potential argument should be set to True. Conversely, setting plot_potential to False directs the method to plot the sensed current.

For introducing noise into the visualization, three distinct types of noise can be applied: gaussian_noise, white_noise, and pink_noise. Activating any of these noise features is achieved by setting the corresponding attribute to True. The introduced noise is composed of random values added to each plot point, sourced from respective probability distributions for Gaussian and white noise, and utilizing functions from the pyplnoise [29] library for pink noise.

Following are three examples of plots for the default double quantum dot device, using the default settings:

```python
# plot the charge stability diagram

# potential, no noise
qdsimulator.plot_charge_stability_diagrams(
    cmapvalue='RdBu', plot_potential=True,
    gaussian_noise=False,
    white_noise=False,
    pink_noise=False)

# current, no noise
qdsimulator.plot_charge_stability_diagrams(
    cmapvalue='RdBu', plot_potential=False,
    gaussian_noise=False,
    white_noise=False,
    pink_noise=False)

# current, noisy
qdsimulator.plot_charge_stability_diagrams(
    cmapvalue='RdBu', plot_potential=False,
    gaussian_noise=True,
    white_noise=True,
    pink_noise=True)
```

The resulting plots are shown in Figure 3.

The noise can further adjusted through specific method attributes: gaussian_noise_params for setting

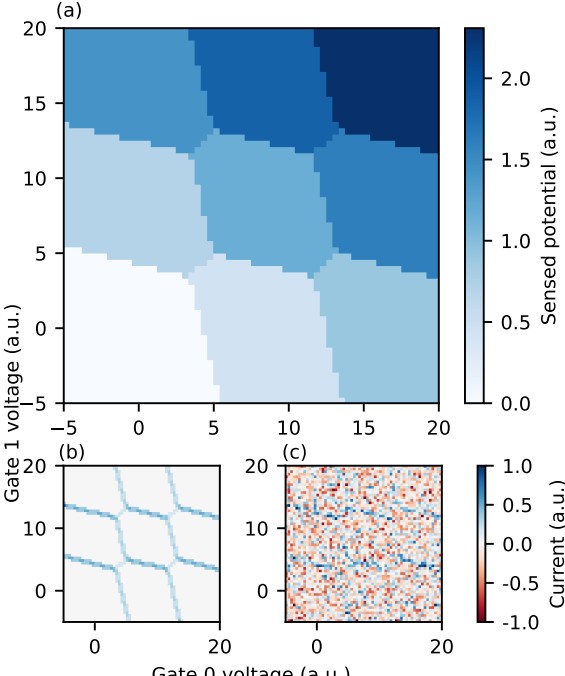

FIG. 3. Simulated charge stability diagrams for a double dot device. In Figure (a) the potential sensed is plotted without noise. In Figure (b) and (c) the gradient is evaluated, therefore plotting the current, with and without noise.

the mean and standard deviation of the distribution, white_noise_params for defining the noise range, and pink_noise_params for adjusting the frequency and amplitude range. For instance, to configure Gaussian noise with a mean of 0.5 and a standard deviation of 0.3, one would set gaussian_noise = True and gaussian_noise_params = [0.5, 0.3].
Lastly, the color scheme of the plot can be personalized by assigning the desired color code to the cmapvalue attribute.

### The crossbar 4x4 shared control device

The crossbar_array_shared_control architecture, as presented in Ref. [6], represents another default configuration accessible to users. In contrast to the one_dimensional_dots_array, this architecture employs a two-dimensional grid layout for dot placement, featuring a shared control system where a single gate can simultaneously influence multiple dots. Such architectures are of particular interest within the academic community for their potential to mitigate the scalability challenge inherent to quantum dot devices. Typically, the number of gates increases linearly with the addition of dots, complicating tuning efforts for expanding device configurations. The shared control approach is considered as an option to lessen this scalability concern.

To simulate a shared-control quantum dot crossbar array, usersfirst create an instance of the QDDevice class and then choose the crossbar_array_shared_control function. Similar to the one_dimensional_dots_array method, specifying the n_dots_side argument as an integer representing the grid's side length automatically configures the device with default parameters. These parameters, may be further customized through either a selection of built-in adjustments (e.g., equal_dots = False, equal_gates = False) or via setter methods for more granular control.

In the following code box, a 4x4 shared-control quantum dot crossbar array is shown. By utilizing the built-in modification functions, we adjust the default settings, subsequently outputting the device's specifications, including the dot-to-dot and dot-to-gate mutual capacitance matrices, and visualizing the device alongside the designated sensor.

```python
# create a quantum dot device object
qddevice = QDDevice()

# crossbar array with shared control
# with 4 dots per side
qddevice.crossbar_array_shared_control(
    n_dots_side=4, equal_dots=False,
    equal_gates=False)

# print the device information
qddevice.print_device_info()

# plot device with sensors and save plot
qddevice.plot_device(
    sensor_locations=[[0,4]],
    sensor_labels=['S0'],
    save_plot_to_filepath='4x4_device.pdf')
```

## Code Output

Device type: crossbar
Number of dots: 16
Number of gates: 7
Physical dot locations:
$[(0, 3), (0, 2), (1, 3), (0, 1), (1, 2),$
$(2, 3), (0, 0), (1, 1), (2, 2), (3, 3), (1, 0), (2, 1), (3, 2),$
$(2, 0), (3, 1), (3, 0)]$
Dot-dot mutual capacitance matrix:

$$\begin{bmatrix}
0.12 & 0.08 & 0.08 & 0.00 & 0.04 & 0.00 & 0.00 & 0.00 & 0.00 & 0.00 & 0.00 & 0.00 & 0.00 & 0.00 & 0.00 & 0.00 \\
0.08 & 0.12 & 0.04 & 0.08 & 0.08 & 0.00 & 0.00 & 0.04 & 0.00 & 0.00 & 0.00 & 0.00 & 0.00 & 0.00 & 0.00 & 0.00 \\
0.08 & 0.04 & 0.12 & 0.00 & 0.08 & 0.08 & 0.00 & 0.00 & 0.04 & 0.00 & 0.00 & 0.00 & 0.00 & 0.00 & 0.00 & 0.00 \\
0.00 & 0.08 & 0.00 & 0.12 & 0.04 & 0.00 & 0.08 & 0.08 & 0.00 & 0.00 & 0.04 & 0.00 & 0.00 & 0.00 & 0.00 & 0.00 \\
0.04 & 0.08 & 0.08 & 0.04 & 0.12 & 0.04 & 0.04 & 0.00 & 0.08 & 0.00 & 0.00 & 0.04 & 0.00 & 0.00 & 0.00 & 0.00 \\
0.00 & 0.00 & 0.08 & 0.00 & 0.04 & 0.11 & 0.00 & 0.00 & 0.08 & 0.08 & 0.00 & 0.00 & 0.04 & 0.00 & 0.00 & 0.00 \\
0.00 & 0.00 & 0.00 & 0.08 & 0.04 & 0.00 & 0.12 & 0.04 & 0.00 & 0.00 & 0.08 & 0.00 & 0.00 & 0.00 & 0.00 & 0.00 \\
0.00 & 0.04 & 0.00 & 0.08 & 0.00 & 0.00 & 0.04 & 0.12 & 0.04 & 0.00 & 0.08 & 0.08 & 0.00 & 0.04 & 0.00 & 0.00 \\
0.00 & 0.00 & 0.04 & 0.00 & 0.08 & 0.08 & 0.00 & 0.04 & 0.12 & 0.04 & 0.00 & 0.08 & 0.08 & 0.00 & 0.04 & 0.00 \\
0.00 & 0.00 & 0.00 & 0.00 & 0.00 & 0.08 & 0.00 & 0.00 & 0.04 & 0.12 & 0.00 & 0.00 & 0.08 & 0.00 & 0.00 & 0.00 \\
0.00 & 0.00 & 0.00 & 0.04 & 0.00 & 0.00 & 0.08 & 0.08 & 0.00 & 0.00 & 0.12 & 0.04 & 0.00 & 0.08 & 0.00 & 0.00 \\
0.00 & 0.00 & 0.00 & 0.00 & 0.04 & 0.00 & 0.00 & 0.08 & 0.08 & 0.00 & 0.04 & 0.11 & 0.04 & 0.08 & 0.08 & 0.04 \\
0.00 & 0.00 & 0.00 & 0.00 & 0.00 & 0.04 & 0.00 & 0.00 & 0.08 & 0.08 & 0.00 & 0.04 & 0.11 & 0.00 & 0.08 & 0.00 \\
0.00 & 0.00 & 0.00 & 0.00 & 0.00 & 0.00 & 0.00 & 0.04 & 0.00 & 0.00 & 0.08 & 0.08 & 0.00 & 0.11 & 0.04 & 0.08 \\
0.00 & 0.00 & 0.00 & 0.00 & 0.00 & 0.00 & 0.00 & 0.00 & 0.04 & 0.00 & 0.00 & 0.08 & 0.08 & 0.04 & 0.12 & 0.08 \\
0.00 & 0.00 & 0.00 & 0.00 & 0.00 & 0.00 & 0.00 & 0.00 & 0.00 & 0.00 & 0.00 & 0.04 & 0.00 & 0.08 & 0.08 & 0.12
\end{bmatrix}$$

Dot-gate mutual capacitance matrix:

$$\begin{bmatrix}
0.15 & 0.00 & 0.00 & 0.00 & 0.00 & 0.00 & 0.00 \\
0.00 & 0.18 & 0.00 & 0.00 & 0.00 & 0.00 & 0.00 \\
0.00 & 0.18 & 0.00 & 0.00 & 0.00 & 0.00 & 0.00 \\
0.00 & 0.00 & 0.12 & 0.00 & 0.00 & 0.00 & 0.00 \\
0.00 & 0.00 & 0.12 & 0.00 & 0.00 & 0.00 & 0.00 \\
0.00 & 0.00 & 0.12 & 0.00 & 0.00 & 0.00 & 0.00 \\
0.00 & 0.00 & 0.00 & 0.13 & 0.00 & 0.00 & 0.00 \\
0.00 & 0.00 & 0.00 & 0.13 & 0.00 & 0.00 & 0.00 \\
0.00 & 0.00 & 0.00 & 0.13 & 0.00 & 0.00 & 0.00 \\
0.00 & 0.00 & 0.00 & 0.13 & 0.00 & 0.00 & 0.00 \\
0.00 & 0.00 & 0.00 & 0.00 & 0.13 & 0.00 & 0.00 \\
0.00 & 0.00 & 0.00 & 0.00 & 0.13 & 0.00 & 0.00 \\
0.00 & 0.00 & 0.00 & 0.00 & 0.13 & 0.00 & 0.00 \\
0.00 & 0.00 & 0.00 & 0.00 & 0.00 & 0.16 & 0.00 \\
0.00 & 0.00 & 0.00 & 0.00 & 0.00 & 0.16 & 0.00 \\
0.00 & 0.00 & 0.00 & 0.00 & 0.00 & 0.00 & 0.13
\end{bmatrix}$$

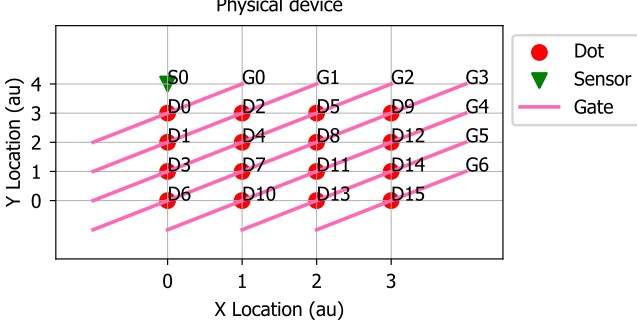

FIG. 4. A schematic plot of the crossbar 4x4 shared control architecture with sensor, which is the output of the `plot_device` method.

In this architecture, gates are arranged diagonally, with the indexing of both dots and gates illustrated in Figure 4.

Similar to the double dot device scenario, simulating the device requires an instance of the `QDSimulator` class. While it's not imperative to create a new simulator instance for each device instance, allowing for the reuse of a single simulator instance, there are situations where dedicating a separate simulator instance to each device may facilitate direct comparisons between simulations. The choice of approach is left to the user's discretion based on their specific requirements.

Following the selection of the physical phenomena to be simulated (either electrons or holes), the user employs the `set_sensor_locations` method to specify the Cartesian coordinates of the sensor(s) involved in the simulation. Subsequently, the `simulate_charge_stability_diagram` function is executed. This involves selecting the device, selecting a solver, setting the voltage ranges, determining the simulation's resolution (noting that higher resolution increases computational demand), and identifying the gates to be scanned.

This example differs from the prior shown in Section due to the presence of additional gates beyond those being actively scanned. To ensure a successful simulation, it is necessary to define the voltage settings for these additional gates. There are two approaches to manage gate voltages in scenarios with more than two gates: uniformly applying voltages using `fixed_voltage` or customizing voltages for each gate via `gates_voltages`. For instance, in a setup with three gates, where Gates 0 and 2 are being scanned and Gate 1 is set to 1.5 volts, this would be represented as `gates_voltages = [None, 1.5, None]`. Alternatively, the same outcome would be achieved by setting `fixed_voltage = 1.5`.

In scenarios with more than three gates, employing `fixed_voltage = 1.5` assigns a uniform voltage of 1.5 to all gates not under scan. In contrast, the `gates_voltages` option permits users to selectively assign specific voltage values to each of the un-scanned gates as per their preference.

It is required to exclusively use either `gates_voltages` or `fixed_voltage` for specifying gate voltages.

In the following code snippet, we assign a uniform voltage of 1 to all gates of secondary interest (specifically Gates 2, 3, 4, 5, and 6).

```
# create a quantum dot simulator object
# simulating electrons
qdsimulator = QDSimulator(simulate=
                          'Electrons')

# set the same sensor locations
qdsimulator.set_sensor_locations([[0, 4]])

# simulate the charge stability diagram
qdsimulator.
    simulate_charge_stability_diagram(
        qd_device=qddevice, solver='MOSEK',
        v_range_x=[-5, 20],
        v_range_y=[-5, 20],
        n_points_per_axis=60,
        scanning_gate_indexes=[0, 1],
        fixed_voltage=1, use_ray=True)
```

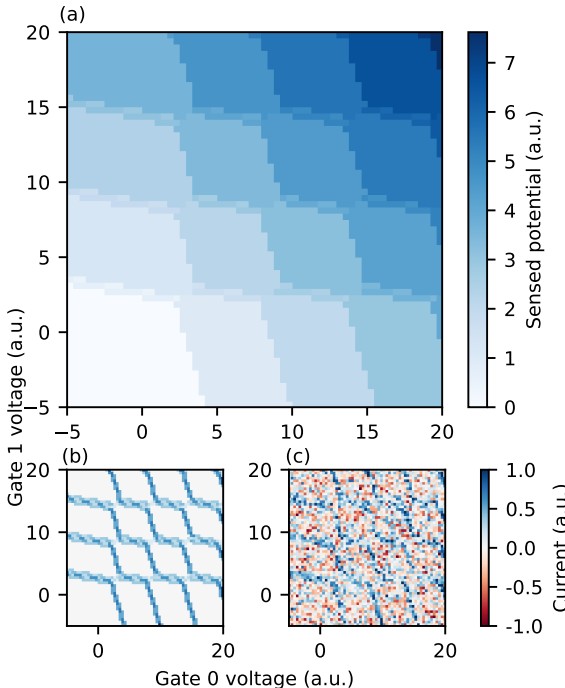

FIG. 5. Simulated charge stability diagrams for a 4x4 shared-control dot device. In Figure (a) the potential sensed is plotted without noise. In Figure (b) and (c) the gradient is evaluated, therefore plotting the current, with and without noise correspondingly.

Employing the identical functions used in the double dot scenario to plot the simulated charge stability diagrams, we generate the subsequent plots shown in Figure 5.

```
# plot the charge stability diagram

# potential, no noise
qdsimulator.plot_charge_stability_diagrams(
    cmapvalue='RdBu', plot_potential=True,
    gaussian_noise=False,
    white_noise=False,
    pink_noise=False)

# current, no noise
qdsimulator.plot_charge_stability_diagrams(
    cmapvalue='RdBu', plot_potential=False,
    gaussian_noise=False,
    white_noise=False,
    pink_noise=False)

# current, noisy
qdsimulator.plot_charge_stability_diagrams(
    cmapvalue='RdBu', plot_potential=False,
    gaussian_noise=True,
    white_noise=True,
    pink_noise=True)
```

### Getting the charge configuration

In order to simplify the labelling of the polytopes, users can take advantage of the `get_charge_configuration` method of the `QDSimulator` class. Users can provide in input a tuple representing the voltage coordinates in the charge stability diagram, and the method will automatically evaluate the closest simulated point and provide the charge configuration at that point. The need to approximate the voltage point comes from the limitation in the granularity of the plot.

An example on how to use the function is shown below.

```
# Let's test the empty region
voltage_point = [0, 0]

# Get the charge configuration of a
# point in the charge stability diagram

print('Charge configuration at chosen
    point', voltage_point, ':')
qdsimulator.get_charge_configuration(
    voltage_point=voltage_point)
```

Charge configuration at chosen point [0, 0] :
Voltage point considered: [0.08474576 0.08474576
1. 1. 1. 1. 1. ]
Charge configuration: [0. 0. 0. 0. 0. 0. 0. 0. 0.
0. 0. 0. 0. 0. 0. 0.]

It is also possible to save the voltage point-charge configuration couple into a variable, as in the following code:

**Python Code**

```python
# Let's test another region
voltage_point = [7, 0]

# Get the charge configuration of a
# point in the charge stability diagram

print('Charge configuration at chosen
    point', voltage_point, ':')
voltage_and_charge_config =
    qdsimulator.get_charge_configuration(
        voltage_point=voltage_point)

print("voltage_and_charge_config =",
    voltage_and_charge_config)
```

**Code Output**

Charge configuration at chosen point [7, 0] :
Voltage point considered: [6.86440678 0.08474576
1. 1. 1. 1. 1. ]
Charge configuration: [1. 0. 0. 0. 0. 0. 0. 0. 0.
0. 0. 0. 0. 0. 0. 0.]
voltage_and_charge_config =
(array([6.86440678, 0.08474576, 1. , 1. , 1. , 1. ,
1. ]), array([1., 0., 0., 0., 0., 0., 0., 0., 0., 0., 0., 0.,
0., 0., 0., 0.]))

### Custom Device Configuration

Users have two main options for configuring custom devices. The first option involves only specifying the locations of the quantum dots. In this case, an individual control system is assumed (one gate per dot), and standard mutual capacitance matrices are derived based on a simple model that considers distance, focusing on first and second nearest neighbors interactions. The second option allows for the further manual specification of the mutual capacitance matrices between dots (dot-to-dot) and between dots and gates (dot-to-gate), thus enabling simulation of complex gate-dot configurations.

Below we show examples of both approaches to custom device simulation.

*Individual control: dot location specification*

First, we consider a scenario where the simulation of a custom device focuses solely on the placement of quantum dots. This can be achieved by creating an instance of the `QDDevice` class and utilizing the `set_custom_dot_locations` method. The order of coordinates tuples directly maps to the dots' indices. The use of `set_custom_dot_locations` triggers an internal function that calculates the mutual capacitance matrices for both dot-to-dot and dot-to-gate interactions. As a result, printing the device information shows these matrices as configured attributes within the class. Similar to the method for the default architectures shown above, minor customizations can be implemented through the boolean parameters `equal_dots`, `equal_gates`, and `crosstalk_strength` for adjusting crosstalk strength. Additionally, the default capacitance value, set at 0.12, can be altered via the `c0` parameter to suit specific requirements.

**Python Code**

```python
# create a quantum dot device object
qddevice = QDDevice()

# set the custom dot locations
qddevice.set_custom_dot_locations([[2, 2],
    [3, 1.5], [4, 2], [1, 1], [5, 1],
    [2, 0], [3, 0.5], [4, 0]],
    equal_dots=False, equal_gates=False,
    crosstalk_strength=0.2, c0=0.12)

# print the device information
qddevice.print_device_info()

# plot the device with sensor
qddevice.plot_device(
    sensor_locations=[[1,2]],
    sensor_labels=['S0'])
```

Device type: custom
Number of dots: 8
Number of gates: 8
Physical dot locations:
$[[2, 2], [3, 1.5], [4, 2], [1, 1], [5, 1], [2, 0], [3, 0.5], [4, 0]]$
Dot-dot mutual capacitance matrix:

$$
\begin{bmatrix}
0.12 & 0.06 & 0.00 & 0.04 & 0.00 & 0.00 & 0.00 & 0.00 \\
0.06 & 0.12 & 0.06 & 0.00 & 0.00 & 0.00 & 0.08 & 0.00 \\
0.00 & 0.06 & 0.12 & 0.00 & 0.04 & 0.00 & 0.00 & 0.00 \\
0.04 & 0.00 & 0.00 & 0.13 & 0.00 & 0.04 & 0.00 & 0.00 \\
0.00 & 0.00 & 0.04 & 0.00 & 0.11 & 0.00 & 0.00 & 0.04 \\
0.00 & 0.00 & 0.00 & 0.04 & 0.00 & 0.12 & 0.06 & 0.00 \\
0.00 & 0.08 & 0.00 & 0.00 & 0.00 & 0.06 & 0.13 & 0.06 \\
0.00 & 0.00 & 0.00 & 0.00 & 0.04 & 0.00 & 0.06 & 0.12
\end{bmatrix}
$$

Dot-gate mutual capacitance matrix:

$$
\begin{bmatrix}
0.14 & 0.01 & 0.00 & 0.01 & 0.00 & 0.00 & 0.00 & 0.00 \\
0.01 & 0.13 & 0.01 & 0.00 & 0.00 & 0.00 & 0.01 & 0.00 \\
0.00 & 0.01 & 0.12 & 0.00 & 0.01 & 0.00 & 0.00 & 0.00 \\
0.01 & 0.00 & 0.00 & 0.12 & 0.00 & 0.01 & 0.00 & 0.00 \\
0.00 & 0.00 & 0.01 & 0.00 & 0.10 & 0.00 & 0.00 & 0.01 \\
0.00 & 0.00 & 0.00 & 0.01 & 0.00 & 0.10 & 0.02 & 0.00 \\
0.00 & 0.01 & 0.00 & 0.00 & 0.00 & 0.02 & 0.14 & 0.01 \\
0.00 & 0.00 & 0.00 & 0.00 & 0.01 & 0.00 & 0.01 & 0.13
\end{bmatrix}
$$

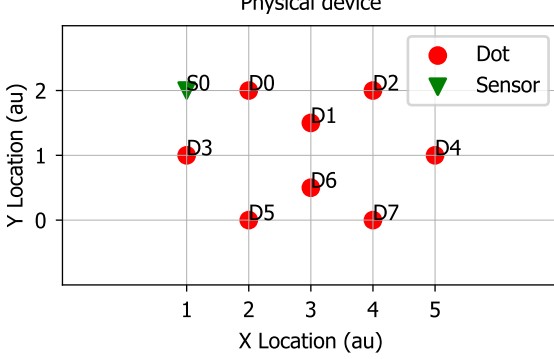

Physical device

FIG. 6. A schematic plot of the custom design with sensor, which is the output of the `plot_device` method.

Figure 6 shows an outline of the custom device, allowing verification of the dot indices against the provided list of coordinates. Currently, a schematic depiction of the gates in custom designs is not available, though such a feature may be incorporated in future releases. In this case each dot is controlled by an individual gate.

For the device simulation itself we simply repeat the procedure defined for the default devices: create a simulator instance, specify the sensor(s) locations, and execute the

`simulate_charge_stability_diagram` method. Note that, despite the system featuring more than two gates, neither the `fixed_voltage` nor the `gate_voltages` parameters are employed. This is due to reliance on the default setting where, in the absence of explicit specifications for these parameters, the system defaults to `fixed_voltage = 0`.

The outcome of the simulation can be plotted like so (output shown in Figure 7):

```python
# create a quantum dot simulator object
qdsimulator = QDSimulator(simulate=
                                'Electrons')

# set the sensor locations
qdsimulator.set_sensor_locations([[1, 2]])

# simulate the charge stability diagram
qdsimulator.
    simulate_charge_stability_diagram(
        qd_device=qddevice, solver='MOSEK',
        v_range_x=[-5, 20],
        v_range_y=[-5, 20],
        n_points_per_axis=60,
        scanning_gate_indexes=[0, 3],
        use_ray=True)

# plot the charge stability diagrams
qdsimulator.plot_charge_stability_diagrams(
    cmapvalue='RdBu', plot_potential=True,
    gaussian_noise=False, white_noise=False,
    pink_noise=False)

qdsimulator.plot_charge_stability_diagrams(
    cmapvalue='RdBu', plot_potential=False,
    gaussian_noise=False, white_noise=False,
    pink_noise=False)

qdsimulator.plot_charge_stability_diagrams(
    cmapvalue='RdBu', plot_potential=False,
    gaussian_noise=True, white_noise=True,
    pink_noise=True)
```

*Shared control: specifying the dot-to-gate mutual capacitance matrix*

In this example, we evolve the previously discussed custom model by transitioning from an individual control system to introducing a custom shared control system, reflected through modifications to the dot-to-gate mutual capacitance matrix.

We adopt a notation that allows us to specify which

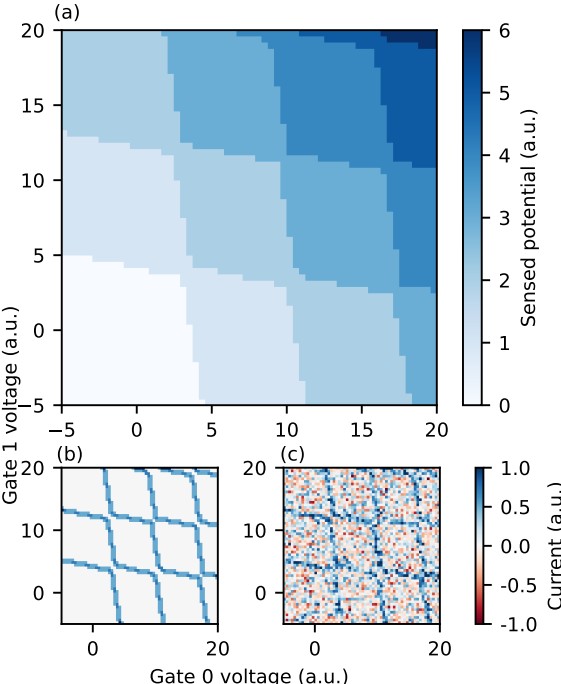

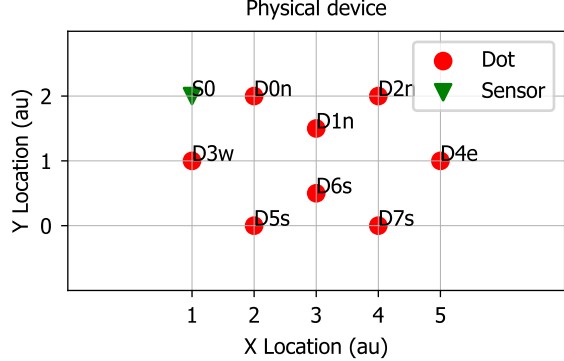

FIG. 8. Another schematic plot of the custom design, in which the labels of the dots have been manually changed to adhere to the new labelling system: 'D' stands for dot, the integer number represents the dot index, the letter represents the gate controlling the dot.

FIG. 7. Simulated charge stability diagrams for a custom design with individual control. In Figure (a) the potential sensed is plotted without noise. In Figure (b) and (c) the gradient is evaluated, therefore plotting the current, respectively with and without noise.

dots are controlled by which gate. Specifically, let's assume we want to use four gates overall to control all dots. We add one of the following four letters: $n, w, e, s$, representing north, west, east, and south, respectively. Dots sharing the same letter are understood to be under the control of the same gate.

Let us illustrate this concept on the concrete example:

```
Python Code

# create a quantum dot device object
qddevice = QDDevice()

# set the custom dot locations
qddevice.set_custom_dot_locations([[2, 2],
    [3, 1.5], [4, 2], [1, 1], [5, 1],
    [2, 0], [3, 0.5], [4, 0]],
    equal_dots=False, equal_gates=False,
    crosstalk_strength=0.2, c0=0.12)

# plot the device with custom labels
# and sensor
qddevice.plot_device(
    sensor_locations=[[1,2]],
    sensor_labels=['S0'],
    custom_dot_labels=['D0n', 'D1n', 'D2n',
        'D3w','D4e', 'D5s','D6s', 'D7s'])
```

In the example above, Dots 0, 1, and 2 are controlled by the northern gate, Dot 3 is governed by the western gate, Dot 4 responds to the eastern gate, while Dots 5, 6, and 7 are controlled by southern gate.

The next step involves encoding this gate-dot architecture into the dot-to-gate mutual capacitance matrix. The dot-to-dot mutual capacitance matrix remains unchanged, still calculated automatically based on the proximity of the dots. For the purpose of matrix notation, we adopt the indexing scheme $[n : 0, w : 1, e : 2, s : 3]$, needed for the construction of an 8x4 matrix representing the dot-to-gate interactions.

This conversion is accomplished through the code snippet below, wherein we employ the `set_dot_gate_mutual_capacitance_matrix` setter method. This allows us to update the dot-to-gate mutual capacitance matrix from the default, automatically generated one, indicative of individual control, to our newly conceptualized matrix that reflects the shared control system.

```
Python Code

dot_gate_matrix = np.array([
    [0.14, 0.00, 0.00, 0.00],
    [0.14, 0.00, 0.00, 0.00],
    [0.14, 0.00, 0.00, 0.00],
    [0.00, 0.13, 0.00, 0.00],
    [0.00, 0.00, 0.12, 0.00],
    [0.00, 0.00, 0.00, 0.15],
    [0.00, 0.00, 0.00, 0.15],
    [0.00, 0.00, 0.00, 0.15]
])

qddevice.
    set_dot_gate_mutual_capacitance_matrix(
        dot_gate_matrix)

qddevice.print_device_info()
```

```
Code Output

Device type: custom
Number of dots: 8
Number of gates: 4
Physical dot locations:
[[2, 2], [3, 1.5], [4, 2], [1, 1], [5, 1], [2, 0], [3, 0.5], [4, 0]]
Dot-dot mutual capacitance matrix:
```

$$\begin{bmatrix} 0.12 & 0.06 & 0.00 & 0.04 & 0.00 & 0.00 & 0.00 & 0.00 \\ 0.06 & 0.12 & 0.06 & 0.00 & 0.00 & 0.00 & 0.08 & 0.00 \\ 0.00 & 0.06 & 0.12 & 0.00 & 0.04 & 0.00 & 0.00 & 0.00 \\ 0.04 & 0.00 & 0.00 & 0.12 & 0.00 & 0.04 & 0.00 & 0.00 \\ 0.00 & 0.00 & 0.04 & 0.00 & 0.12 & 0.00 & 0.00 & 0.04 \\ 0.00 & 0.00 & 0.00 & 0.04 & 0.00 & 0.12 & 0.06 & 0.00 \\ 0.00 & 0.08 & 0.00 & 0.00 & 0.00 & 0.06 & 0.12 & 0.06 \\ 0.00 & 0.00 & 0.00 & 0.00 & 0.04 & 0.00 & 0.06 & 0.12 \end{bmatrix}$$

Dot-gate mutual capacitance matrix:

$$\begin{bmatrix} 0.14 & 0.00 & 0.00 & 0.00 \\ 0.14 & 0.00 & 0.00 & 0.00 \\ 0.14 & 0.00 & 0.00 & 0.00 \\ 0.00 & 0.13 & 0.00 & 0.00 \\ 0.00 & 0.00 & 0.12 & 0.00 \\ 0.00 & 0.00 & 0.00 & 0.15 \\ 0.00 & 0.00 & 0.00 & 0.15 \\ 0.00 & 0.00 & 0.00 & 0.15 \end{bmatrix}$$

Finally, we proceed to simulate and plot the charge stability diagrams in Figure 9.

```
Python Code

# create a quantum dot simulator object
qdsimulator = QDSimulator(simulate=
                            'Electrons')

# set the sensor locations
qdsimulator.set_sensor_locations([[2, 1]])

# simulate the charge stability diagram
qdsimulator.
    simulate_charge_stability_diagram(
        qd_device=qddevice, solver='MOSEK',
        v_range_x=[-5, 20],
        v_range_y=[-5, 20],
        n_points_per_axis=60,
        scanning_gate_indexes=[0, 3],
        use_ray=True)

# plot the charge stability diagrams
qdsimulator.plot_charge_stability_diagrams(
    cmapvalue='RdBu',plot_potential=True,
    gaussian_noise=False, white_noise=False,
    pink_noise=False)

qdsimulator.plot_charge_stability_diagrams(
    cmapvalue='RdBu',plot_potential=False,
    gaussian_noise=False, white_noise=False,
    pink_noise=False)

qdsimulator.plot_charge_stability_diagrams(
    cmapvalue='RdBu', plot_potential=False,
    gaussian_noise=True, white_noise=True,
    pink_noise=True)
```

## PERFORMANCE EVALUATION AND CONSTRAINTS

QDsim package is designed to equip the scientific community with a user-friendly tool for simulating charge stability diagrams of arbitrary architectures. Our primary goal is to create an accessible codebase, allowing researchers to efficiently simulate charge stability diagrams for extensive quantum dot arrays with a minimal setup and little preliminary knowledge. The ultimate objective of QDsim package is to introduce a rapid and efficient tool for generating comprehensive datasets for subsequent use in machine learning model training.

Among comparable available tools, the QTT package [30] stands out, although it primarily addresses highly specialized experimental scenarios. QTT is geared more towards analysis and measurements within precise experimental frameworks, rendering it somewhat challenging for beginners and limiting its applicability to a nar-

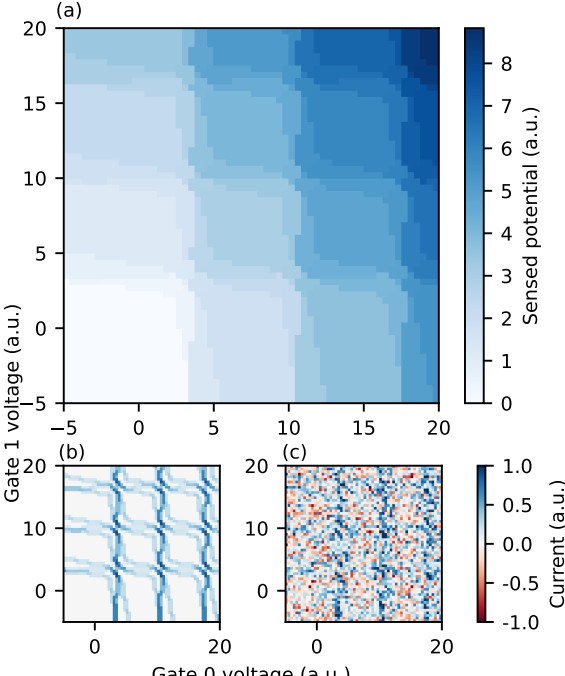

FIG. 9. Simulated charge stability diagrams for a custom design with shared control. In Figure (a) the potential sensed is plotted without noise. In Figure (b) and (c) the gradient is evaluated, therefore plotting the current, respectively with and without noise.

row range of devices (e.g., double dots, in-line 6 dot array, triple dot, square dot) all under individual control schemes. This specificity in focus is the primary reason a detailed comparison with `QTT` has not been pursued, as the two packages cater to distinct needs and applications. Furthermore, very recently a new package named `SimCATS` [31] has been published. It aims to achieve a maximally realistic description of CSD especially with regards to sensing and noise implementations. While having a different focus, it caters to the need of realistic noise and could be a promising addition to our approximated solution. Furthermore, our work is complementary to a simultaneously published package called `QDarts` [32], which leverages a polytope-finding algorithm to efficiently simulate and locate charge transitions in the presence of tunnel couplings, non-constant charging energy and realistic noise. However due higher computational complexity, it focuses on smaller dot arrays (approximately 10 dots).

Focusing on the strengths of our package, `QDsim` distinguishes itself with its speed and capability to simulate extensive arrays. To quantify these advantages, all simulations and time measurements discussed herein were conducted on an Apple M2 chip equipped with 16 GB of memory.

Speed optimization in the package is influenced by two main factors: the device architecture size (i.e. number of dots and gates), reflected by the matrix dimensions, which offers limited parallelization opportunities, and the plot granularity, which can benefit from advanced parallelization techniques.

Here we compare across several configurations, including double dots, in-line 6 dots, and shared control arrays of sizes 4x4, 6x6, and 8x8.

The comparison results are illustrated in Figure 10.

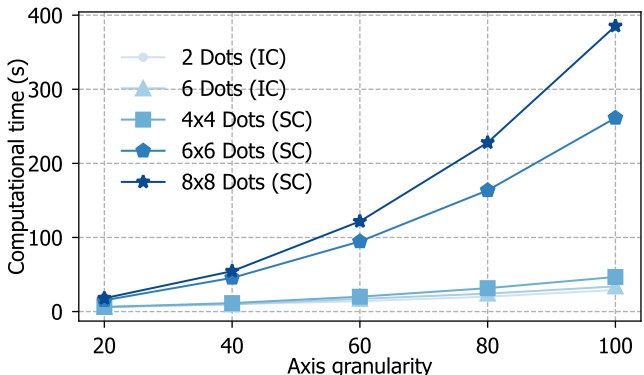

FIG. 10. Comparing the computational time (in seconds) to simulate a specific architecture, with respect to the granularity of the plot's axis. The number of points evaluated for the plot is therefore the x-axis numbers squared. From the plot we can see how the computational time increases with increasing number of plot points, but also with the size of the device, i.e. more dots require more computational time for a given axis granularity. A 64-dot device with shared control system can be simulated with good granularity on a laptop in less than 4 minutes.

Granularity requirements will vary based on the desired plot quality and voltage space size; naturally, broader voltage ranges would likely necessitate higher granularity levels. For instance, in our example section, a voltage range of 25 and a granularity level of 60 yielded satisfactory plots in under a minute, highlighting the package's emphasis on speed and ease of use.

When considering limitations, we identify two main categories: physical and functional constraints. On the physical side, it is crucial to acknowledge that the simulations are based on electrostatics and do not account for quantum mechanical effects. This limitation impacts the simulations' realism, potentially affecting the precision in modeling actual devices. However, the package's primary goal is not to perfectly replicate physical systems but rather to generate datasets for machine learning applications, where models can be initially trained on sufficiently accurate simulated data and later fine-tuned with real experimental data.

From a functional perspective, potential enhancements could include improved visualizations of gate configurations, optimization of parallel processing routines for increased speed, and the development of a graphical user

interface (GUI). This GUI could allow users to intuitively position dots and gates, with the software automatically suggesting starting points for mutual capacitance values between dots and gates.

## CONCLUSION

The development of this package was driven by the ambition to offer the scientific community a highly accessible and user-friendly tool, specifically designed to streamline the generation of charge stability diagrams for extensive quantum dot arrays. With a focus on efficiency and speed, this package aims to significantly reduce the time and complexity traditionally associated with such simulations. By simplifying the process for both students and professionals, whether in theoretical or experimental domains, the package opens up new avenues for exploration and discovery in the field of quantum computing and nanotechnology. Furthermore, it lays the groundwork for the creation of comprehensive datasets, essential for the advancement of machine learning applications within this sphere.

As we look forward to contributions from the community, enriching the package with more sophisticated models, our ultimate hope is that it becomes a cornerstone for innovation, fostering advancements that leverage both computational simulations and machine learning to unravel the complexities of quantum systems.

## ACKNOWLEDGEMENTS

The authors acknowledge fruitful discussions with Francesco Borsoi, Brennan Undseth, Maia Rigot, and Menno Veldhorst.

*Funding information* This research was supported by the European Union's Horizon Europe programme under the Grant Agreement 101069515 – IGNITE. This publication is part of the project Engineered Topological Quantum Networks (with Project No. VI.Veni.212.278) of the research program NWO Talent Programme Veni Science domain 2021 which is financed by the Dutch Research Council (NWO).

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
