# Peer review of "QDsim: A user-friendly toolbox for simulating large-scale quantum dot devices"

_SciPost Physics Codebases, doi:SciPost Phys. Codebases 46 (2025) , SciPost Phys. Codebases 46-r1.1 (2025)_

## Round 1 · Referee Report · Dietmar Weinmann (Referee 1) · 2024-6-20

Strengths

1- The code allows to treat devices with many quantum dots 2- The computation of the stability diagram is rather fast

Weaknesses

1- The code is based on approximations as the constant-interaction model that limit the accuracy in certain regimes 2- When the capacitances are not user-provided, their determination from the sample geometry is unclear. 3- It is unclear how some quantities are computed, for example the "current", while important underlying parameters like the tunnel couplings and the temperature are not among the input data of the code.

Report

The authors present a software package to compute charge stability diagrams for devices containing a large number of coupled quantum dots. As a motivation, they mention the possible production of large datasets for the training of AI-based automatic tuning of quantum dot arrays for applications in quantum technology.
The presented software is based on the so-called constant-interaction model which approximates the effect of the electron-electron interactions in the device through capacitances. Within this approximation, the charge stability diagrams are found by identifying the charge distributions with integer electron numbers on each dot that minimize the free energy of the device which is expressed in terms of the capacitance matrix, the applied gate voltages, and the numbers of charges on the quantum dots.
The proposed package is described in the paper, with example applications, and it could be useful for the efficient simulation of the model devices within the approximations it is based on. The gitlab repository contains installation instructions and a tutorial section with examples that should allow users to use the software. However, I couldn't find a full-fledged detailed documentation of the code.
It seems to me that the paper thereby more or less meets the criteria of SciPost Physics Codebases. Before the paper is considered for publication, I would nevertheless suggest that the authors take into account the remarks in the list of requested changes.

Requested changes

1- It is not obvious that quantum dots, and in particular small ones with a low electron number, are well described by the constant-interaction model. Errors in the simulation may have fatal consequences in the proposed application of the software since the quality of an AI depends crucially on the quality of the training data. The authors should therefore discuss the limitations of accuracy induced by their approximations and describe the regimes where their software can be expected to provide acceptable results. A benchmarking demonstration of the accuracy through the comparison with more precise codes could be appropriate.

2- The quantum dot device is described by its capacitance matrix which includes the dot-to-dot capacitances as well as the dot-to-gate capacitances. Those capacitances can be provided by the user, but they can also be determined by the software based on the physical dot locations. The latter functionality seems crucial for users who want to simulate a given device. Unfortunately, the paper does not describe how exactly the capacitances are determined and how the detailed properties of the sample such as regions with different dielectric constants are taken into account. A precise simulation would however need a rather thorough treatment of the electrostatic environment (see for example Chatzikyriakou et al., Phys. Rev. Research 4, 043163 (2022)). The authors should describe how their software determines the capacitance matrix from the device geometry, and discuss the approximations involved. A demonstration of the accuracy through the comparison with more precise codes for the case of a small device or with an experimental device might be appropriate.

3- While some randomness in the capacitance matrix may be useful to describe a large ensemble of similar samples, it is not obvious to me why it is useful to add different kinds of noise to the output data. Maybe the authors want to comment on their motivation to have such a feature in the software.

4- The lower panels of Figs. 3, 5, 7, 9 show current data. However, it is not clear where this current flows, where the source and the drain are located, and what allows electrons to move in the sample. Capacitive coupling is not enough as an input for a current calculation, it is the tunnel couplings between dots that are needed to have current. Moreover, the current in quantum dot devices usually depends strongly on temperature, which is also not among the input data. It should be explained how the current is calculated within the described software. Moreover, in the captions of those Figures it is said that (b) shows the current with noise and (c) without noise. From looking at the plots, one has the opposite impression. Are the subfigures (b) and (c) inverted?

5- The definition of "e" below eq. (7) as the elementary charge with sign +1 for holes and -1 for electrons is confusing. It is not clear to me what convention is used exactly to describe the charges.

Recommendation

Ask for minor revision

  • validity: high
  • significance: high
  • originality: high
  • clarity: good
  • formatting: excellent
  • grammar: excellent

Author:  Valentina Gualtieri  on 2024-08-02  [id 4673]

(in reply to Report 1 by Dietmar Weinmann on 2024-06-20)

Thank you for your valuable feedback. We appreciate the opportunity to clarify the limitations and the scope of applicability of our software, QDsim. We also wanted to share that a documentation is now available at https://qdsim.readthedocs.io/ Attached you can find an updated version of our submission in which we highlighted in red the changes made.

Here is our response addressing your questions:

1- It is not obvious that quantum dots, and in particular small ones with a low electron number, are well described by the constant-interaction model. Errors in the simulation may have fatal consequences in the proposed application of the software since the quality of an AI depends crucially on the quality of the training data. The authors should therefore discuss the limitations of accuracy induced by their approximations and describe the regimes where their software can be expected to provide acceptable results. A benchmarking demonstration of the accuracy through the comparison with more precise codes could be appropriate.

Our primary goal is not to precisely model the intricate physics of quantum dots, but rather to provide a scalable and efficient method for generating large datasets as those are a key bottleneck in many driven algorithms for control and tuning of quantum devices. These sizable datasets are crucial for pretraining machine learning (ML) models, which can then be fine-tuned with more precise data. The constant-interaction model serves this purpose by qualitatively reproducing key features of charge stability diagrams (CSDs). The constant-interaction model is rooted in classical electrostatics and does not incorporate quantum transport phenomena, such as tunneling effects. Spin interactions are also neglected, which can be significant in small quantum dots with low electron numbers. While the model provides clear charge sections and charge transition lines, it does not capture the full complexity of the quantum mechanical interactions within the dots. The model does not account for temperature variations, which can influence the electronic properties and stability of quantum dots. Despite these limitations, the constant-interaction model effectively captures the qualitative behavior of charge stability diagrams. The features it reproduces, such as charge sections and transition lines, are the primary aspects used for device control and tuning in contemporary experiments. This makes the model a valuable tool for generating preliminary datasets for machine learning applications. Machine learning models require extensive datasets for effective training. Generating such datasets experimentally can be time-consuming and expensive. Our approach provides a scalable solution to this problem. Models pre-trained on our simulated data can achieve a good initial performance. These models can then be fine-tuned with more precise experimental data, significantly improving their accuracy and robustness. We acknowledge that models trained solely on our simulated data may not generalize perfectly to all experimental conditions. However, they provide a solid foundation that can be refined with targeted experimental datasets constructed either using measured data labeled by a human operator or generated with more sophisticated numerical techniques that include quantum effects. Our software aims to streamline the initial stages of ML model development by providing a fast and scalable method for generating training data. While the constant-interaction model has its limitations, it serves as a useful tool for pretraining models, which can then be fine-tuned with more precise and specific data. We believe this approach strikes a balance between efficiency and accuracy, enabling rapid advancements in quantum dot research and applications. We added a benchmark with the other open-source quantum dot array packages (see Fig. 11 in the resubmitted manuscript). We also extended our discussion of full quantum simulation and discussed qualitative comparison with these works (Section: Performance Evaluation and Constraints).

We understand the importance of accurately describing the methods used to determine the capacitance matrix and the need to account for the detailed properties of the sample, such as regions with different dielectric constants. Here is our response to address your concerns: 2- “The quantum dot device is described by its capacitance matrix which includes the dot-to-dot capacitances as well as the dot-to-gate capacitances. Those capacitances can be provided by the user, but they can also be determined by the software based on the physical dot locations. The latter functionality seems crucial for users who want to simulate a given device. Unfortunately, the paper does not describe how exactly the capacitances are determined and how the detailed properties of the sample such as regions with different dielectric constants are taken into account. A precise simulation would however need a rather thorough treatment of the electrostatic environment (see for example Chatzikyriakou et al., Phys. Rev. Research 4, 043163 (2022)). The authors should describe how their software determines the capacitance matrix from the device geometry, and discuss the approximations involved. A demonstration of the accuracy through the comparison with more precise codes for the case of a small device or with an experimental device might be appropriate.” We appreciate the reviewer's emphasis on the importance of accurately determining the capacitance matrix from the device geometry and the electrostatic environment. We acknowledge that a detailed treatment of the electrostatic environment, including regions with different dielectric constants, is crucial for precise simulations. In this package however we did not aim at realism and precision, but mostly at a high-level representation of the electrostatic environment. In our package, the capacitance matrices are determined based on physical dot locations using electrostatics. Let’s break down how we evaluated the dot-to-dot and the dot-to-gate capacitance matrices. The dot-to-dot capacitance matrix is calculated using a distance-based model. For each pair of dots, the mutual capacitance is determined based on their physical separation. The diagonal elements of the matrix represent the self-capacitance of each dot, which is initially set to a user-defined value (c0). Off-diagonal elements are calculated based on the inverse square of the distance between dots to approximate the capacitive coupling. Similar to the dot-to-dot capacitance, the dot-to-gate capacitance matrix is initially set based on a distance-based model. For each dot-gate pair, the mutual capacitance is determined based on their physical separation (where the gate controlling the corresponding dot is considered as placed in the same coordinates of the dot it controls), with an additional factor to account for crosstalk effects between neighboring gates. Random noise can be added to simulate variations in gate capacitances and to introduce crosstalk. The precise methods used can be found in our open source gitlab repo, under the _QuantumDotDevice.py module. The method used are called evaluate_dot_gate_mutual_capacitance_matrix And evaluate_dot_dot_mutual_capacitance_matrix . A slightly different evaluation is done for the crossbar array architecture: the information can be found in the method evaluate_dot_gate_mutual_capacitance_matrix_crossbar. We added detailed description of how we calculate the capacitances as well as the discussion of the limitation of our capacitance estimation approach as well as references to more sophisticated capacitance determination methods (in QDDevice class description).

  1. While some randomness in the capacitance matrix may be useful to describe a large ensemble of similar samples, it is not obvious to me why it is useful to add different kinds of noise to the output data. Maybe the authors want to comment on their motivation to have such a feature in the software. We appreciate the reviewer's observation regarding the addition of different kinds of noise to the output data. Our motivation for including this feature in QDsim is rooted in the practical realities of experimental quantum dot devices and the specific needs of machine learning applications. In real-world quantum dot experiments, noise is an inevitable part of the measurement process. Sources of noise include thermal fluctuations, electronic interference, and imperfections in the fabrication of the devices. By incorporating noise into the simulated data, we aim to create more realistic training datasets for machine learning models. These models are often used for automating the tuning and control of quantum dot devices, and training them on noise-free data could lead to suboptimal performance in real experimental conditions. Furthermore, machine learning models trained on data with various types and levels of noise tend to be more robust and generalize better to unseen data. This is because the models learn to identify and manage the underlying signal amidst the noise, rather than overfitting to idealized, noise-free conditions. (e.g. — Page 273, Neural Smithing: Supervised Learning in Feedforward Artificial Neural Networks, 1999.) By adding Gaussian, white, and pink noise, we expose the models to a wide range of potential experimental scenarios, enhancing their ability to perform reliably under different conditions. Introducing noise into the simulation allows researchers to test and validate the resilience of their algorithms and control strategies. By evaluating performance under noisy conditions, researchers can gain insights into the limits of their approaches and make necessary adjustments. Users have the flexibility to add these types of noise individually or in combination, based on their specific simulation needs. The noise parameters can be adjusted to control the mean, standard deviation, and amplitude, providing a customizable approach to noise modeling. By incorporating noise into the output data, we aim to bridge the gap between idealized simulations and real-world experimental conditions. This approach enhances the robustness and generalizability of machine learning models trained on our simulated data, ultimately contributing to more reliable and effective quantum dot device control strategies. We included a detailed explanation of the motivations and benefits of adding noise to the output data in the revised manuscript, addressing the reviewer's concerns and providing clarity for future users of the software.

  2. The lower panels of Figs. 3, 5, 7, 9 show current data. However, it is not clear where this current flows, where the source and the drain are located, and what allows electrons to move in the sample. Capacitive coupling is not enough as an input for a current calculation, it is the tunnel couplings between dots that are needed to have current. Moreover, the current in quantum dot devices usually depends strongly on temperature, which is also not among the input data. It should be explained how the current is calculated within the described software. Moreover, in the captions of those Figures it is said that (b) shows the current with noise and (c) without noise. From looking at the plots, one has the opposite impression. Are the subfigures (b) and (c) inverted?

We appreciate the reviewer's helpful comments regarding the current calculations and the presentation of our figures. Here, we provide a detailed response to address these concerns.

Current: In our simulations, the current is not explicitly calculated based on physical tunnel couplings or temperature dependencies. Instead, we take the gradient of the sensed potential matrix to estimate the current. This approach is a simplified method to provide a qualitative visualization of how the sensed potential changes. We acknowledge that this method does not account for tunnel couplings between dots, source and drain locations, or temperature effects. As a result, the calculated "current" is a qualitative measure rather than a precise physical quantity. For precise current calculations, detailed information about the tunnel couplings, temperature, and specific source-drain configurations would be necessary. Incorporating these factors could be a direction for future development. We added a detailed description of how the current is calculated. Clarification of Figures and Captions: We apologize for the confusion caused by the labeling of subfigures in Figs. 3, 5, 7, and 9. Upon review, it appears that the subfigures (b) and (c) were indeed inverted. We corrected the captions in the revised manuscript to accurately reflect the content of each subfigure. We also enhanced the captions to include a brief explanation of how the current is derived from the potential gradient. This will provide readers with a clearer understanding of what the figures represent. We also added a section in the manuscript that explicitly describes how the current is calculated within the software. This section will detail the use of the potential gradient and discuss the assumptions and limitations of this method. 5.- The definition of "e" below eq. (7) as the elementary charge with sign +1 for holes and -1 for electrons is confusing. It is not clear to me what convention is used exactly to describe the charges. We understand that the definition of "e" and the conventions used to describe the charges may be confusing. We provided a more precise and clear explanation to ensure that readers can easily understand the charge conventions employed in our manuscript. We updated the definition introducing a neutral charge symbol q, where q is the unit charge of the carriers. For electrons, q = -e and for holes, q = +e , with e being the absolute value of the elementary charge (e = 1.602 \times 10^{-19} $ Coulombs).

We hope this response addresses the concerns raised, and we are committed to making the necessary revisions to ensure the manuscript provides a comprehensive overview of the software's capabilities and limitations. Thank you once again for your constructive feedback.

The authors

Attachment:

QMAI_getuning_qdsim_arxiv-3.pdf

---

## Round 1 · Referee Report · Anonymous (Referee 2) · 2024-7-8

Strengths

Fast (as demonstrated in Fig. 10) generation of charge stability diagrams, even for large arrays of quantum dots.

Weaknesses

The package offers computation of various physical quantities, including current (see Fig. 9) and its noise, but details of how such computation is performed and under what kind of approximations are lacking.

Report

The manuscript presents new software for computation of properties of quantum dot arrays. The paper meets criteria for publication in SciPost Physics Codebases, but some improvements are needed.

Requested changes

Discussion of limitations of approximations (such as constant-interaction model, see abstract "package currently does not support quantum effects beyond the constant interaction model") used in modeling of quantum dot arrays are missing. In general, when developing any code, it is advisable to benchmark its results against small systems which have exact analytical or numerical (from another code) solution. Such benchmarking is missing, so it should be added in the revised manuscript.

Recommendation

Ask for minor revision

  • validity: good
  • significance: good
  • originality: good
  • clarity: ok
  • formatting: good
  • grammar: excellent

Author:  Valentina Gualtieri  on 2024-08-02  [id 4674]

(in reply to Report 2 on 2024-07-08)

We thank the referee for highlighting the challenges of appropriate benchmarking. Since our reply contains figures, the referee can read it here: https://docs.google.com/document/d/1kNAzu1Y-6HUtChYRg4E95UV4BiJL9m6mhqxWKUPZm68/edit?usp=sharing

I've attached an updated pdf version of our submission with the changes highlighted in red.

Thank you for your time,
The authors

Attachment:

QMAI_getuning_qdsim_arxiv-3_2FNju4u.pdf

---

## Editorial Decision

published